# Citrate and malonate increase microbial activity and alter microbial community composition in uncontaminated and diesel contaminated soil microcosms

Belinda C. Martin[1], Suman J. George[2], Charles A. Price[1], Esmaeil Shahsavari[3], Andrew S. Ball[3], Mark Tibbett[4], Megan H. Ryan[1]

[1]School of Plant Biology, The University of Western Australia, Crawley WA, 6009, Australia
[2]School of Earth and Environment, The University of Western Australia, Crawley WA, 6009, Australia
[3]Centre for Environmental Sustainability and Bioremediation, School of Applied Sciences, RMIT University, Melbourne VIC, 3082, Australia
[4]Centre for Agri-Environmental Research, School of Agriculture Policy and Development, University of Reading, Berkshire, RG6 6AR, United Kingdom

*Correspondence to*: Belinda C. Martin (belinda.martin@research.uwa.edu.au)

**Abstract.** Petroleum hydrocarbons (PHCs) are among the most prevalent sources of environmental contamination. It has been hypothesized that plant root exudation of low molecular weight organic acid anions (carboxylates) may aid degradation of PHCs by stimulating heterotrophic microbial activity. To test their potential implication in bioremediation, we applied two commonly-exuded carboxylates (citrate and malonate) to uncontaminated and diesel contaminated microcosms (10,000 mg kg$^{-1}$; aged 40 days) and determined their impact on the microbial community and PHC degradation. Every 48 hours for 18 days, soil received 5 µmol g$^{-1}$ of i) citrate, ii) malonate, iii) citrate + malonate or iv) water. Microbial activity was measured daily as the flux of $CO_2$. After 18 days, changes in the microbial community were assessed by CLPP and 16S rRNA bacterial community profiles determined by DGGE. Saturated PHCs remaining in the soil were assessed by GC-MS. Cumulative soil respiration increased four- to six-fold with the addition of carboxylates, while diesel contamination resulted in a small, but similar, increase across all carboxylate treatments. The addition of carboxylates resulted in distinct changes to the microbial community in both contaminated and uncontaminated soils, but only a small increase in the biodegradation of saturated PHCs as measured by the *n*-C17: pristane biomarker. We conclude that while the addition of citrate and malonate had little direct effect on biodegradation of saturated hydrocarbons present in diesel, their effect on the microbial community leads us to suggest further studies using a variety of soils and organic acids and linked to *in situ* studies of plants to investigate the role of carboxylates in microbial community dynamics.

*Key words:* CLPP; carboxylates; organic acids; petroleum hydrocarbons; phytoremediation; rhizosphere

## 1. Introduction

Contamination of soils by petroleum hydrocarbons (PHCs) can impair soil function and pose serious risks to human and ecosystem health (Park and Park, 2011; Ramadass et al., 2015). A novel approach for remediating surface soils contaminated with PHCs is to grow plants in order to benefit from their stimulating effect on rhizosphere microorganisms (bioremediation) (Anderson et al., 1993; Ma et al., 2010). Plant roots provide favorable conditions for rhizosphere microorganisms largely through the exudation of substrates that allow increased growth and activity (*e.g.* amino acids, carbohydrates and organic acids). It has been speculated the increase in microbial growth and activity that is driven by root exudates accelerates the rate of PHC biodegradation within the rhizosphere (Anderson et al., 1993; Nie et al., 2011; Martin et al., 2014; Phillips et al., 2012; Shahsavari et al., 2015). For instance, root exudates have been linked to enhanced PHC degradation for plants grown *in situ* (Joner and Leyval, 2003; Gao et al., 2011) and in batch experiments in which exudates collected from plant roots were added to contaminated soils (Miya and Firestone, 2001; Yoshitomi and Shann, 2001; Xie et al., 2012).

Low molecular weight (LMW; molecular weight <500 MW) aliphatic organic acid anions (carboxylates) comprise a significant proportion of root exudate profiles (Jones, 1998; Ryan et al., 2001). Carboxylates may act as biostimulating agents in the degradation of PHCs through the provision of a labile carbon source that supports increased microbial growth and activity. When added to soils, carboxylates are rapidly degraded by soil microorganisms, with up to 80% mineralized to $CO_2$ within 24 hours, depending upon the carboxylate added and the soil type (Evans, 1998; Ström et al., 2001; Hashimoto, 2007; Oburger et al., 2009; Fujii et al., 2010; Ryan et al., 2012). In addition, carboxylates have the capacity to enhance soil phosphorus supply and, hence, microbial growth and activity through phosphate desorption either due to anion exchange or their ability to chelate to metal cations (*e.g.* $Al^{3+}$, $Fe^{3+}$ and $Ca^{2+}$) (Jones and Darrah, 1994; Ryan et al., 2001; Shane and Lambers, 2005;). Carboxylates may also function to increase the bioavailability of PHCs by promoting their desorption from the soil matrix (Ling et al., 2009; An et al., 2010, 2011; Gao et al., 2010; Keiluweit et al., 2015;), although this would depend on soil type and may require higher carboxylate concentrations than reportedly occur in soils (Ryan et al., 2001).

Previous work which aimed to assess the importance of root exudates for bioremediation has often employed model systems using a combination of rhizosphere components (e.g. carboxylates in combination with carbohydrates and amino acids) (Joner and Leyval, 2003; Miya and Firestone, 2001; Xie et al., 2012). Such an approach makes it impossible to determine which exudate component is causing the greatest shift in microbial community structure and function, and is therefore of most importance for degradation of PHCs. Carboxylate release can differ greatly among plant species with only some species, notably those species possessing cluster roots, known to release high concentrations of carboxylates (Dinkelaker et al., 1989; Grierson, 1992; Jones, 1998; Strobel, 2001). To determine whether plant species with high carboxylate exudation should be prioritised when selecting plants to screen for ability to enhance PHC degradation, we devised a simple test system using two commonly exuded carboxylates.

It should also be noted that while many studies have investigated the role of carboxylates in plant nutrient acquisition and aluminium detoxification (Pang et al., 2010; Pearse et al., 2006; Shane and Lambers, 2005), the impact of carboxylates on the soil microbial community has received considerably less attention. Additionally, whilst studies have examined the microbial biodegradation of carboxylates in contrasting soil types (Fujii et al., 2010, 2012; Hashimoto, 2007; Jones and Darrah, 1994), soil horizons (Evans, 1998; van Hees et al., 2002), and with variations in the initial pH of the organic acid applied (Ström et al., 2005), as far as we are aware, there are no studies examining in detail the biodegradation of carboxylates, and their effect on the microbial community, in soils contaminated with PHCs.

The objective of this study was, therefore, to determine the impact of two carboxylates, citrate and malonate, on microbial activity and community structure in uncontaminated and diesel-contaminated microcosms. We selected these carboxylates as they are among two of the most commonly exuded carboxylates (Jones, 1998; Kidd et al., 2016). In the diesel-contaminated soil, the impact of carboxylate addition on the degradation of PHCs was also examined. We hypothesized that: i) addition of citrate and malonate would enhance microbial activity (soil respiration); ii) addition of diesel and carboxylates would lead to shifts in the microbial community structure and function; and iii) addition of citrate and malonate would enhance degradation of PHCs.

## 2. Materials and methods

### 2.1 Soil characteristics

A grey loamy sand with 5% clay, (grey chromosol; ASC, arenosol; WRB), previously uncontaminated by PHCs, was collected from the top mineral horizon (0–10 cm) of a pasture on a dairy farm 90 km south of Perth, Western Australia (32°45'31.16S, 115°49'33.88E). The soil was passed through a 2 mm mesh, stored in cloth bags and kept field moist at 4 °C until used. The soil contained 16 mg kg$^{-1}$ of $NH_4$-N and 121 mg kg$^{-1}$of $NO_3$-N (Searle, 1984), 52 mg kg$^{-1}$ of bicarbonate-extractable P and 61 mg kg$^{-1}$ of bicarbonate-extractable K (Colwell, 1963), 88 mg kg$^{-1}$ of KCl-extractable S (Blair et al., 1991) and 3.3% organic C (Walkley and Black, 1934); pH $_{(CaCl2)}$ was 4.6 and electrical conductivity was 34 mS m$^{-1}$. Soil water retention was determined after equilibrating saturated soil at a series of gas pressures (–0.1, –10, –33 and –100 kPa).

### 2.2 Experimental design

Previously uncontaminated soil was housed in gas-tight glass microcosms (385 mL). Soil in half the microcosms was spiked with 10,000 mg kg$^{-1}$ of diesel (obtained from a Caltex Inc. commercial bowser). The concentration of diesel (10,000 mg kg$^{-1}$) was selected to make results comparable to previous studies on diesel bioremediation (Bento et al., 2005; Boopathy, 2004; Seklemova et al., 2001; Tesar et al., 2002). Sterile deionized water was added to the microcosms to 40% of water holding capacity to negate moisture limitation (Tibbett et al., 2011). The soil was mixed thoroughly and incubated in the dark at 24 °C

for 40 days. To prevent the build-up of volatiles and to allow gas exchange, microcosms were regularly aerated under a laminar flow and watered to weight to maintain the water holding capacity at 40%.

Stock solutions of citric and malonic acid solutions were prepared and the pH adjusted to 4.5 with KOH to ensure that the organic acids were predominantly in anionic form (Jones and Darrah, 1995). Every 48 hours, both diesel-contaminated and uncontaminated soils received either: i) citrate (5 µmol $g^{-1}$ soil), ii) malonate (5 µmol $g^{-1}$ soil), iii) citrate + malonate (2.5 µmol $g^{-1}$ soil per carboxylate), or iv) sterile deionized water (Table S1). Carboxylates were added for 18 days, equating to a total of 45 µmol $g^{-1}$ soil. These concentrations were chosen to reflect those likely to occur in the rhizosphere (Jones, 1998; Neumann and Romheld, 1999; Shane and Lambers, 2005). There were four replicates in each treatment. Microcosms were incubated in the dark at 24 °C when not in use.

**2.3 Soil respiration**

Soil respiration was measured daily for 18 days on all microcosms using an infrared gas analyzer (Series 225 Gas Analyzer, Hoddesdon, UK) as previously described (Clegg et al., 1978). Measurement of $CO_2$ respiration is a common indicator for measuring changes in microbial activity and has been shown to directly correlate with the extent of PHC degradation (Baptista et al., 2005; Greenwood et al., 2009). Briefly, a sample of headspace gas (1 mL) was collected with a syringe from each microcosm and injected into the gas analyzer sample line. All samples were calibrated against a $CO_2$ standard (5.08% v/v). Following $CO_2$ analysis, the headspace of each microcosm was refreshed by airing with a fan for 20 seconds.

**2.4.1 Post incubation measures**

The experiment was halted after 18 days of carboxylate addition. Subsamples from three replicates of each treatment combination were removed for measurement of soil $pH_{(CaCl2)}$, as described previously, and determination of the community level physiological profile (commonly referred to as 'CLPP'). Microbial community analysis using 16S rRNA bacterial community profiles determined by denaturing gradient gel electrophoresis (DGGE) was performed on two replicates of the diesel-contaminated treatments and the water control in the uncontaminated treatment. Saturated hydrocarbon analysis was performed on three replicates of the diesel-contaminated treatment.

**2.4.2 Community level physiological profile**

CLPPs were determined using the MicroResp™ micro-plate system as described by Campbell et al. (2003). MicroResp is a rapid, culture-independent technique which allows assessment of active members of the microbial community based on their use of various carbon substrates (Campbell et al., 2003). Thirty-one known rhizosphere or root-exuded carbon substrates (Campbell et al., 1997; Degens and Harris, 1997; Banning et al., 2012) were dissolved in water and added to deep-well plates to make a final concentration of 30 mg C g soil water$^{-1}$ (Table S2). The addition of each carbon substrate was replicated three times randomly throughout the plate. Soil (~0.23 g) was added to each well and the plates were then sealed and covered with

a detection plate containing 150 µL Cresol Red agar (1% Noble Agar, 20 µg mL$^{-1}$ Cresol Red, 240 mM KCl and 4 mM NaHCO$_3$). The assembled system was secured with clamps and incubated in the dark for 4 hours at 24 °C. Carbon dioxide was measured based upon the colorimetric reaction that occurs in the detection plate due to the change in pH as CO$_2$ reacts with bicarbonate. The absorbance for each detection plate was measured before (pre-assay) and after (post-assay) at 590 nm in a
micro-plate reader (Multiskan, Thermo Scientific) using Skan-it software (version 2.2, Thermo Scientific). Post-assay absorbance values were normalized against pre-assay absorbance and converted to headspace CO$_2$ using a calibration curve obtained with detachable 'combi strip' dye wells (8 wells each) exposed to known volumes of standard CO$_2$ in sealed test tubes (first flushed with N$_2$) for 4 hours at 24 °C.

### 2.4.3 Microbial community analysis

Soil DNA was extracted using a PowerSoil$^{®}$ DNA Isolation Kit (MoBio laboratories, Inc. USA) according to the manufacturer's guidelines. The soil bacterial community was assessed by PCR using universal primers 341F-GC and 917R on 16S rRNA genes. All PCR products were examined using 1.5% agarose gel electrophoresis prior to DGGE analysis (Simons et al., 2012). DGGE was carried out using the Universal Mutation Detection System (BioRad) with a 6% urea-formamide denaturant gradient polyacrylamide (40–60% denaturing gradient). The gel was run at 60 °C and 60 V for 18 hours, silver
stained as described previously (Simons et al., 2012) and scanned using an Epson V700 scanner.

### 2.4.4 Saturated petroleum hydrocarbon analysis

Accelerated Solvent Extraction (ASE 200 $^{®}$) was used to extract remaining saturated PHCs from the soil. Soil (5 g) was mixed with diatomaceous earth (1:1), packed into a 33 mL extraction cell and extracted with 50% acetone and 50% hexane at 200 °C and 1500 psi, with a static time of 5 min, a flush volume of 60% and a purge time of 60 seconds. A soil sample freshly spiked
with diesel fuel (10,000 mg kg$^{-1}$ of diesel) was also extracted at this time. The saturated PHCs were then separated using small column chromatography and eluted with n-pentane (Bastow et al., 2007). N-tetracosane was added as an internal standard. An aromatic fraction was subsequently eluted with *n*-pentane/dichloromethane (7:3 v/v), but for technical reasons could not be analyzed. The analysis of the saturated fraction was performed with an Agilent 7890A GC system using an HP-5 MS Agilent column (length 30 m, internal diameter 250 µm, film thickness 0.25 µm). Soil freshly spiked with diesel was also analyzed.
The oven was programmed for 40 °C for 3 minutes then increased by 15 °C min$^{-1}$ to 320 °C with a total run time of 26 minutes and a helium carrier gas flow of 1.0 mL min$^{-1}$. MS conditions were ionization energy of 70 eV and source temperature of 230 °C. Product identifications were based on the retention profile and correlation to the internal standard. Peak areas were integrated using Agilent Enhanced ChemStation software (E.02.02.1431). Biodegradation was assessed by two commonly-used biomarker ratios, *n*-C17:pristane and *n*-C18:phytane, that were first normalized by the internal standard (Greenwood et
al., 2008).

**2.5 Statistical analysis**

A two-way analysis of variance (ANOVA) was performed using Genstat version 14.1 (Lawes Agricultural Trust, Rothamsted Experimental Station, Harpenden, UK) to determine the effect of carboxylate addition (water control, malonate, citrate + malonate, citrate) and diesel contamination (absent, present) upon cumulative soil respiration and soil pH. A one-way ANOVA was used to determine the effect of carboxylate addition on the biodegradation of saturated PHCs in the diesel-contaminated soil. The least significant difference at $P = 0.05$ is reported for significant main effects or interactions. Data were log transformed when required to satisfy the linear model assumptions.

CLPP data were first standardized by subtracting the control substrate ($CO_2$ response from cells containing only water) within each treatment and transformed to achieve normality using a fourth root transformation. A resemblance matrix (Table S3) was created using Euclidean distance and differences among *a priori* defined groups were examined by permutational multivariate analysis of variance (PERMANOVA) using Primer (version 6.1, PRIMER-E) and presented in an ordination using principal coordinates (PCO) (Anderson, 2001; Anderson and Willis, 2003). Pairwise comparisons were performed to determine whether CLPP data differed significantly among treatment combinations. Canonical analysis of principal coordinates (CAP) was also performed (Anderson and Willis, 2003). In addition, CLPP data were divided into four main classes (amino acids [n = 6 substrates], aromatics [n = 6 substrates], carbohydrates [n = 3 substrates] and carboxylates [n = 12 substrates]: see Table S2.). Each class was then analyzed separately with a three-way ANOVA to determine the effect of carboxylate addition, diesel contamination and carbon substrate on $CO_2$ evolution during the four hours of the CLPP assay.

The DGGE gel was subject to analysis with Phoretix 1D software (Phoretix Ltd, UK). The similarities between the communities based on presence/absence of bands were expressed using the unweighted pair group method using arithmetic averages (UPGMA).

**3. Results**

**3.1 Microbial activity**

Cumulative $CO_2$ evolved over the 18-day incubation was increased by carboxylate addition ($P < 0.001$) and diesel contamination ($P < 0.001$), but there was no interaction between these two factors (Fig. 1). The addition of carboxylates increased cumulative $CO_2$ up to six-fold compared with the water control. Addition of malonate resulted in lower $CO_2$ evolution than addition of citrate + malonate or citrate (estimated means: malonate = 1.02, citrate + malonate = 1.33, citrate = 1.45, $LSD_{0.05} = 0.09$). Diesel contamination resulted in a small increase in cumulative $CO_2$ evolved in all carboxylate treatments (estimated means: uncontaminated = 0.90, diesel contamination = 1.11, $LSD_{0.05} = 0.06$).

(Fig. 1)

While respiration was only measured every 48 hours in this experiment, it was recorded more frequently in a preliminary experiment (Fig. S1). In this preliminary experiment, by one hour after addition, the respiration rate had increased up to 16-fold for citrate and four-fold for malonate; it then halved by three hours after addition, remained steady until 24 hours after addition and then declined until reaching control levels by 74 hours.

### 3.2 Microbial community structure and function

### 3.2.1 Community level physiological profiles

Principal coordinates analysis (PCO) revealed a separation based on the microbial CLPPs among soils that received carboxylates and the water controls (Fig. 2; Table S3). The initial three PCO axes together accounted for 67.5% of the variation in the data (Fig. 2A). PERMANOVA of the CLPP data found an effect of carboxylate addition ($P_{perm} = 0.004$) with malonate ($P_{perm} = 0.002$) and citrate ($P_{perm} = 0.001$), but not citrate + malonate, as compared to the water control (Fig. 2A). There was no effect of diesel contamination and no interaction between carboxylate addition and diesel contamination.

(Fig. 2)

Canonical analysis of the CLPP data, where the number of treatments and replicate structure is specified into *a priori* defined groups, showed a clear difference among the three treatments when carboxylates were added and the water controls, again irrespective of whether soils were diesel contaminated or not (Fig. 2B).

To further understand these utilization patterns, the 31 carbon substrates were divided into four main carbon groups (amino acids, aromatics, carbohydrates, carboxylates; Table S2) and for each group the effect of carboxylate addition, diesel contamination and individual substrate was examined using a three-way ANOVA (Table 1; Fig. 3). The carboxylate treatment affected the utilization of all four classes of compounds. Utilization of amino acids, aromatics and carboxylates was greatly enhanced in soil which had carboxylates applied previously, while for carbohydrates there was a complex three-way interaction (results not shown). Diesel contamination affected the utilization of amino acids and carboxylates only. For amino acids, diesel contamination reduced the utilization only in the citrate + malonate treatment. For the carboxylates, utilization was always lower in the diesel-contaminated soil.

(Fig. 3)
(Table 1)

### 3.2.2 Microbial community analysis

DGGE of 16S rRNA gene bacterial community profiles showed clear differences among treatments (Fig. 4). Bacterial communities receiving only water distinctly differed between uncontaminated and diesel-contaminated soils. However, a greater differentiation occurred with the addition of carboxylates with the soils that received carboxylates (all diesel-contaminated) strongly grouped compared with the water controls from the uncontaminated and diesel-contaminated soils. Within the diesel-contaminated soil, the citrate + malonate treatment was more closely related to citrate than to malonate. The water control in the diesel-contaminated soil showed two distinct bands that were much less distinct in the carboxylate treatments and in the water control in the uncontaminated soil.

(Fig. 4)

### 3.3 Soil pH

For soil pH there was an interaction between the carboxylate treatment and the diesel-contamination treatment ($P = 0.001$) (Table 2). This interaction was a result of the diesel contamination increasing pH only in the water control and malonate treatment. The carboxylate treatment caused pH to increase in the order of citrate $\geq$ citrate + malonate > malonate > water control.

(Table 2)

### 3.4 Degradation of saturated petroleum hydrocarbons

In the diesel-contaminated soil, the GC-MS chromatograms from the carboxylate addition treatments and the water control showed a reduction in saturated alkanes when compared to soil freshly spiked with diesel (Table 3). Total ion chromatograms of the carboxylate addition treatments and the water control showed a loss of small chain alkanes and an increasing prominence of unresolved complex mixture compared to the freshly spiked soil (data not shown). The biomarker *n*-C17:pristane was slightly, but significantly, reduced with the addition of carboxylates and the *n*-C18:phytane biomarker showed a similar trend, albeit not significant (Table 3). There was no significant difference in peak area of any of the individual saturated hydrocarbons with carboxylate addition (Table 3).

(Table 3)

## 4. Discussion

### 4.1 Citrate and malonate enhanced microbial activity in uncontaminated and diesel-contaminated soils

In the uncontaminated and diesel-contaminated soils, cumulative $CO_2$ evolution increased greatly in response to 18 days of repeated carboxylate addition compared to the water control. While some carboxylates may have been mineralized abiotically

(such as may occur in the presence of Mn oxides), the relative contribution of this pathway is known to be minor (Jones et al., 1996). We, therefore, assume that in our study the increase in $CO_2$ evolution with addition of carboxylates was a result of carboxylate mineralization by the soil microbes. Thus, our first hypothesis that carboxylates enhance microbial activity was supported.

The rapid increase in soil respiration with the addition of citrate is in accordance with previous studies and is indicative of the ability of the soil microbial community to quickly mineralize carboxylates (Evans, 1998; van Hees et al., 2002; Jones et al., 1996). It implies a fast turnover rate of carboxylates in soils, which is consistent with van Hees et al. (2005) who estimated a mean residence time for carboxylates of between 2 and 70 hours in the top mineral horizons, equating to a turnover of up to ten times a day (Van Hees et al., 2005).

There was a clear difference in the ability of citrate and malonate to stimulate microbial activity; with citrate addition (either by itself or in combination with malonate) resulting in the highest cumulative $CO_2$ evolution in diesel-contaminated and uncontaminated soils. Interestingly, malonate has been labeled a potential inhibitor of microbial respiration due to its inhibition of succinate dehydrogenase, a key enzyme of the citric acid cycle (Ikuma and Bonner, 1967; Li and Copeland, 2000; Phillips

et al., 2012). However, our results show that malonate greatly increases soil respiration. This finding is consistent with that of Oburger et al. (2009) who found malonate did not inhibit microbial activity when added to a range of soil types in solutions also containing malate, citrate and oxalate.

### 4.2 Citrate and malonate impact soil microbial physiological profiles and community structure

The addition of citrate and malonate altered the microbial community in uncontaminated and diesel-contaminated soils,

leading to the development of distinct communities as defined by the community level physiological profiles (CLPP) and the 16S rRNA bacterial community profiles (DGGE). The changes induced by the addition of carboxylates were complex and included, not surprisingly, an increased ability to use carboxylates, as well as amino acids and aromatics. Interestingly, the DGGE of the 16S rRNA bacterial community profiles showed two distinct bands in the diesel-contaminated water control which were less distinct when carboxylates had been added. Unfortunately, these bands were not sequenced so we do not know

the identity of these microbes and whether they are known PHC degraders. However, the community shifts suggest that citrate and malonate have the potential to play a key role in shaping microbial community structure even in diesel-contaminated soils.

Future studies incorporating 16S amplicon sequence analysis as well as functional gene analysis would help to uncover some of the key microbial players in these carboxylate-induced community shifts.

Soil pH also increased with carboxylate addition and it should be noted that pH can have a large influence on the composition of soil microbial communities (Fierer and Jackson, 2006; Lauber et al., 2009; Rousk et al., 2010). It is, therefore, highly likely that changes in soil pH are confounded with the effect of carboxylate addition on microbial community structure. Carboxylate addition has been shown to increase pH in the absence of plants and this has been attributed to microbial-mediated decarboxylation, as during decarboxylation a proton is consumed according to the following reaction: $R\text{-}CO\text{-}COO^- + H^+ \rightarrow R\text{-}CHO + CO_2$ (Yan et al., 1996; Rukshana et al., 2012; Keiluweit et al., 2015). The fact that soil pH was highest in the treatment with the greatest cumulative $CO_2$ evolution (citrate) is consistent with this scenario. However, the increase in soil pH may also be due to the reaction of carboxylate ions with free $H^+$ in the soil solution, ammonification of organic nitrogen or $NO_3^-$ uptake by microorganisms (Hinsinger et al., 2003). Soil acidification poses a potential impediment to biodegradation as PHCs are optimally degraded within a near neutral pH (Dibble and Bartha, 1979). However, carboxylate exudation in plants has only been weakly correlated to rhizosphere acidification (Pang et al., 2010; Pearse et al., 2006), thus highlighting one of the limitations of applying model systems to real rhizospheres and underlining the need for future *in-situ* studies with high carboxylate exuding plants.

**4.3 Citrate and malonate did not significantly enhance the degradation of saturated PHCs over the course of 18 days**

Loss of saturated hydrocarbons occurred in all treatments, including the water control, as shown by the reduction in saturated hydrocarbons compared to soil freshly spiked with diesel fuel. A large majority of these losses likely occurred before the addition of carboxylates due to initial volatilization of small chain hydrocarbons as the jars were vented. Carboxylate addition over 18 days did not have a major impact on the degradation of saturated hydrocarbons, as we only detected a slight change in one of the biodegradation biomarkers (*n*-C17:pristane) with the addition of carboxylates.

Carboxylates appear to have selected for microbes that are more stimulated by carboxylates than PHCs, thus leading to no major degradation of PHC's compared to the water control. Although no major significant differences were found, there was a trend of decreasing PHC abundance with addition of carboxylates over the time course of 18 days. Whether this trend would continue given a longer experimental time period is unknown. However, given phytoremediation trials often last longer than one year (Vervaeke et al., 2003; Gurska et al., 2009;) the time frame of the current study may not have been long enough to detect significant declines.

Citrate, malate and oxalate have been shown to increase the desorption of phenanthrene and pyrene from various soils, however relatively high (up to 1000 mM *cf.* up to 600 µM in rhizosphere) concentrations were used (Ling et al., 2009; An et al., 2010,

2011; Gao et al., 2010). The concentration of carboxylates applied in this study was chosen to be representative of the rhizosphere (e.g. 600 uM) and may have been too low to have caused any significant desorption effects on the PHCs.

## 4.4 Limitations of this study and future research

The hypothesized links between enhanced PHC degradation in the rhizosphere and plant roots extend beyond the effects of a single carbon compound. Plant roots exude a wide array of compounds (*e.g.* amino acids, exo-enzymes, secondary metabolites) which, when released in localized areas, may act synergistically to both directly and/or indirectly enhance microbial PHC degradation (Kuiper et al., 2004; Martin et al., 2014). Therefore, the role of carboxylates in enhancing microbial PHC biodegradation may be greater in the presence of other root compounds (*e.g.* secondary metabolites) that enable a more diverse microbial community to be present. Additionally, in our study the experimental design and the techniques employed to reduce sample variability (*e.g.* root-free soil, constant rate of carboxylate supply, and sifting and mixing soil) are far from what would be experienced in a rhizosphere environment and so these results cannot be uncritically extrapolated to bioremediation studies performed *in situ*. Nonetheless, as our findings show that citrate and malonate can have a significant effect on microbial community structure in soil microcosms, further investigation focusing on plant-microbial interactions with high carboxylate exuding plants is warranted. These studies should incorporate next generation 16S sequencing technologies as well as functional gene analysis to uncover some of the key microbial players in these carboxylate-induced community shifts. Also, there is much more to learn about what might occur under different edaphic environments, with different types of PHCs and when other carboxylates or combinations of carboxylates are present.

## 5. Conclusion

The key outcome of this preliminary study is that citrate and malonate can stimulate microbial activity and alter microbial community structure in both uncontaminated and diesel-contaminated soils. While we found no strong evidence to suggest that addition of citrate and malonate enhanced degradation of saturated hydrocarbons during the short (18 day) timeframe of this model system study, their impact on the microbial community leads us to suggest that further investigation of their role in plant-microbial relations is warranted using modified and more diverse experimental systems linked to *in-situ* studies of plants growing in contaminated soils.

## 6. Acknowledgements

This work was funded by an Australian Research Council linkage project (LP110201130) in partnership with Horizon Power, ChemCentre and Environmental Earth Sciences. CAP was supported by a Discovery Early Career Research Award from the Australian Research Council. Many thanks to Natalie Joyce, Daniel Ramsden, Daniel Kidd and Navjot Kaur for their technical assistance.

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

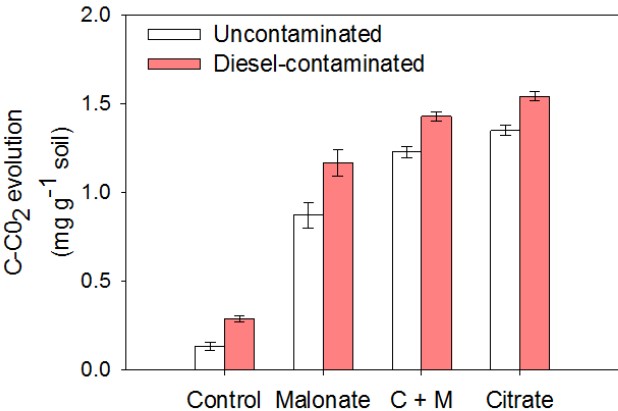

**Figure 1: Cumulative CO₂ evolved over the 18-day incubation in diesel-contaminated and uncontaminated soils which received water (control) or 5 µmol g⁻¹ soil of citrate, malonate, or citrate + malonate every 48 hours. Values are means ± SE (n=4). There was an effect of carboxylate addition ($P<0.001$) and diesel-contamination ($P<0.001$), but no significant interaction.**

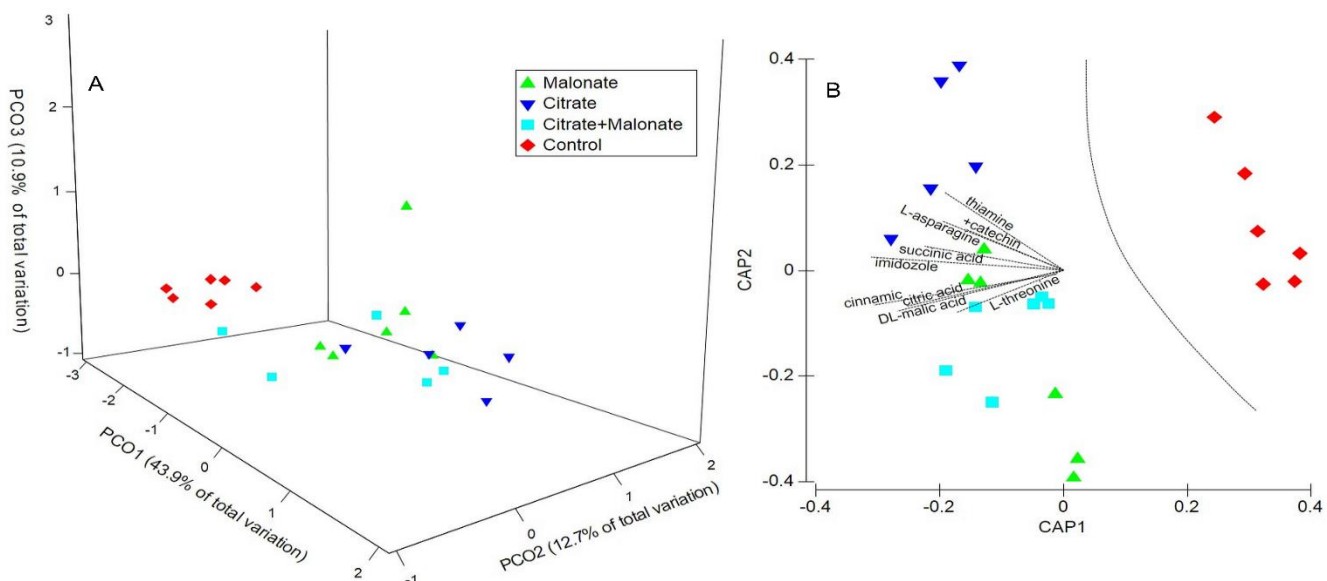

**Figure 2: (a) Principal co-ordinate analysis ordination (PCO) and (b) canonical analysis of principal co-ordinates (CAP) of CLPP**
10 **on soil after 18 days of incubation for diesel-contaminated and uncontaminated soils which received water (control) or 5 µmol g⁻¹ soil of citrate, malonate, or citrate + malonate every 48 hours. As there was no effect of diesel-contamination there is no distinction made between the three replicates of each carboxylate treatment from diesel-contaminated soil and the three replicates from uncontaminated soil. The curve in Fig. 2 (b) represents a significant difference ($P_{Perm}=0.004$ based on PERMANOVA) between treatments receiving carboxylates and the water control.**

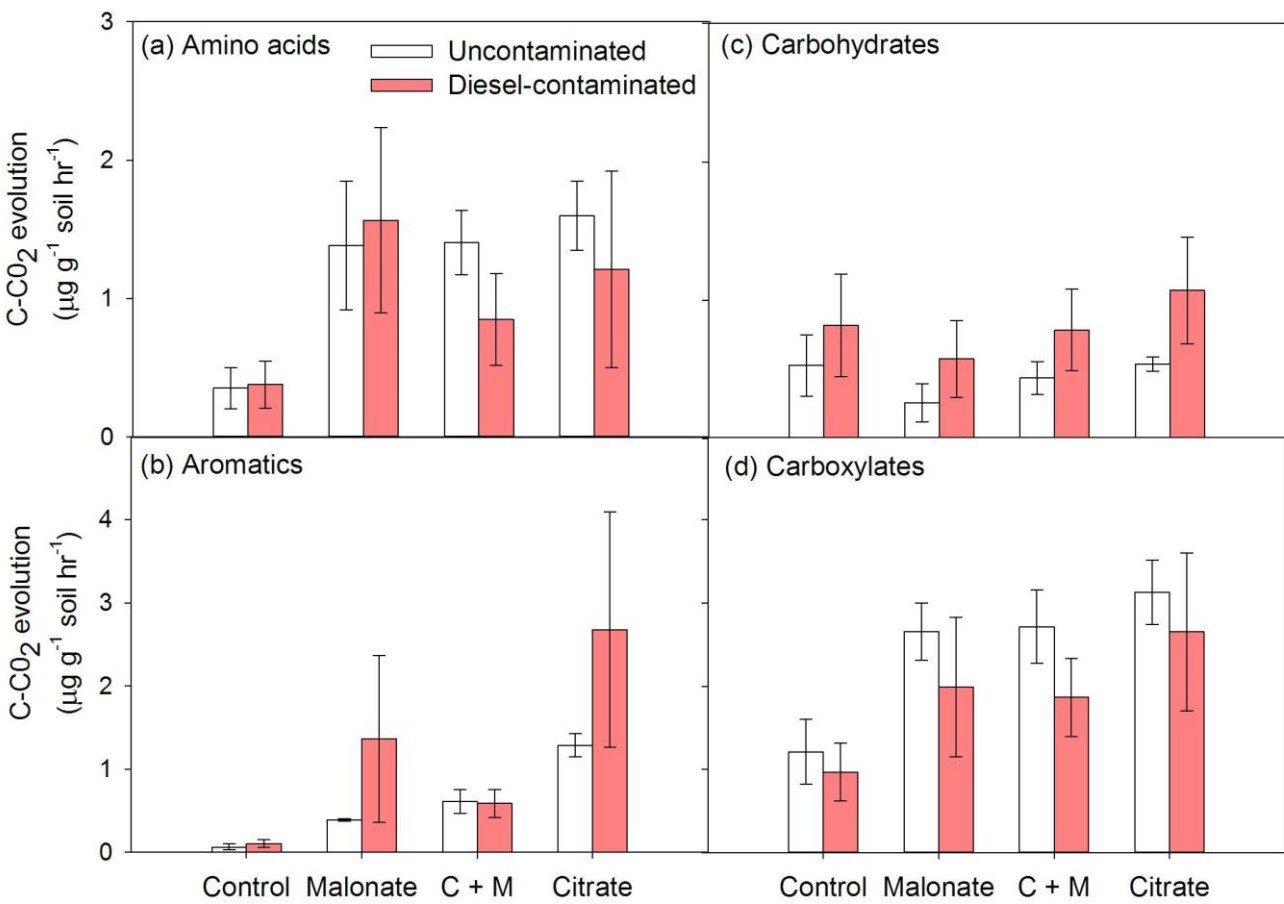

**Figure 3: Estimated means for the interaction of carboxylate treatment and diesel contamination for CLPP on soil after 18 days of incubation for diesel-contaminated and uncontaminated soils which received water or 5 µmol g$^{-1}$ soil of malonate, citrate + malonate or citrate every 48 hours and was mixed with 27 substrates from four main classes (amino acids [n=6 substrates], aromatics [n=6 substrates], carboxylates [n=12 substrates], carbohydrates [n=3 substrates]. Values are means ± SEM (n=3). Statistical outcomes are presented in Table 3.**

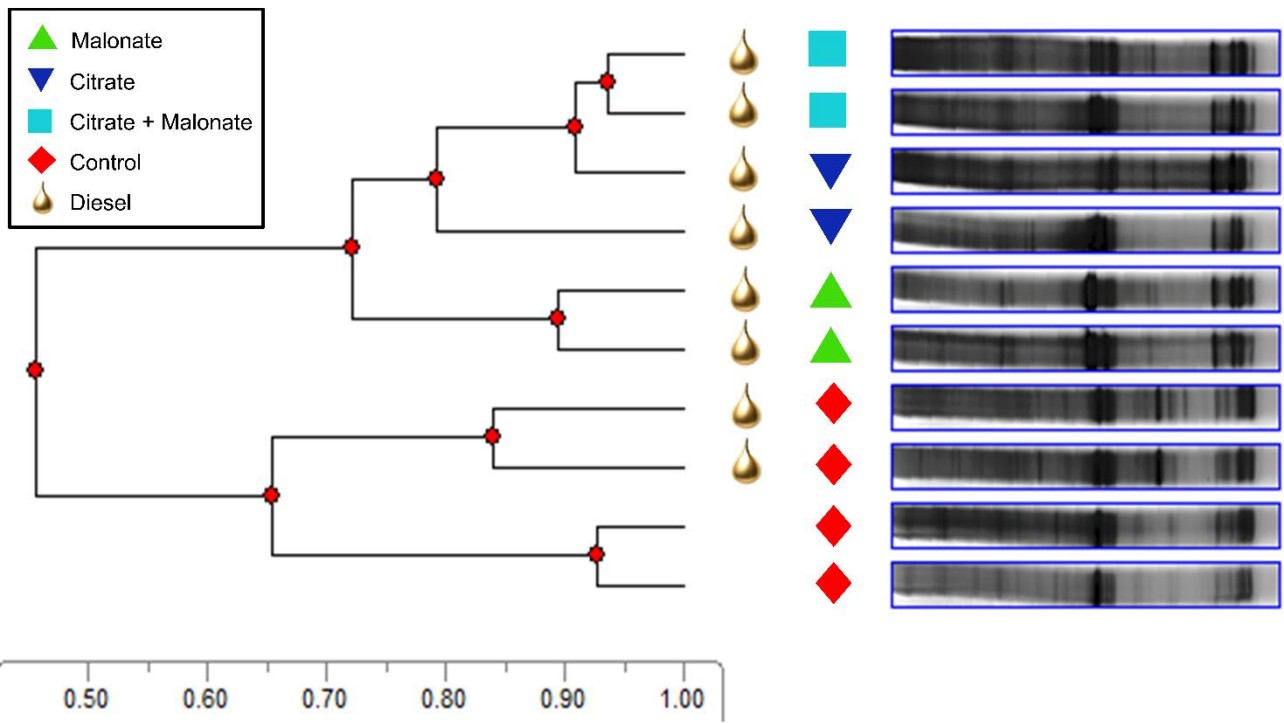

**Figure 4: UPGMA dendrogram of 16s rRNA DGGE bacterial community profiles after 18 days of incubation in diesel-contaminated and uncontaminated soil which received water (control) and diesel-contaminated soil that received 5 µmol g$^{-1}$ soil of malonate, citrate + malonate or citrate every 48 hours. The scale represents the similarity coefficient.**

**Table 1. Outcomes of a three-way analysis of variance on the community level physiological profiles (CLPPs) after an 18-day incubation of diesel-contaminated and uncontaminated soils which received water or 5 µmol g$^{-1}$ soil of malonate, citrate + malonate or citrate every 48 hours.**

| Substrate class | Carboxylate addition (C) | Diesel contamination (D) | Substrate (S) | C × D | C × S | D × S | C × D ×S |
|---|---|---|---|---|---|---|---|
| Amino acids (n = 6) | <0.001 | 0.02 | 0.046 | 0.013 | n.s.* | n.s. | n.s. |
| Aromatics (n = 6) | <0.001 | n.s. | <0.001 | n.s. | n.s. | n.s. | n.s. |
| Carbohydrates (n = 3) | n.s. | n.s. | <0.001 | n.s. | n.s. | n.s. | 0.041 |
| Carboxylates (n = 12) | <0.001 | 0.001 | <0.001 | n.s. | n.s. | n.s. | n.s. |

* n.s. = not significant

**Table 2. Soil pH$_{(CaCl2)}$ after an 18-day incubation of diesel-contaminated and uncontaminated soils which received water or 5 µmol g$^{-1}$ soil of malonate, citrate + malonate or citrate every 48 hours. Values are means ± SE (n = 3). There was an interaction between carboxylate addition and diesel contamination ($P$ = 0.001, LSD$_{0.05}$ = 0.055). Initial soil pH was 4.6 and carboxylate solution pH was 4.5.**

|  | Uncontaminated soil | Diesel-contaminated soil |
|---|---|---|
| Control | 4.47 ± 0.01 | 4.67 ± 0.02 |
| Malonate | 6.64 ± 0.01 | 6.78 ± 0.01 |
| Citrate + malonate | 7.18 ± 0.01 | 7.23 ± 0.02 |
| Citrate | 7.29 ± 0.01 | 7.31 ± 0.04 |

**Table 3. Saturated hydrocarbons in a soil freshly spiked with diesel and after an 18-days incubation of diesel-contaminated soils which received water or 5 µmol g⁻¹ soil of malonate, citrate + malonate, or citrate every 48 hours. Values are standardized mean peak area ± SE (n = 3).**

| | Hexadecane ($n$-C16) | Heptadecane ($n$-C17) | Octadecane ($n$-C18) | Norpristane | Pristane | Phytane | $n$-C17:Pr | $n$-C18:Ph | Pr:Ph |
|---|---|---|---|---|---|---|---|---|---|
| Freshly spiked soil | 8.521 | 8.526 | 7.590 | 2.828 | 4.363 | 2.767 | 1.954 | 2.744 | 1.577 |
| Treatment | | | | | | | | | |
| Control | 0.274 ± 0.021 | 0.408 ± 0.047 | 0.276 ± 0.035 | 0.763 ± 0.098 | 1.823 ± 0.267 | 1.376 ± 0.149 | 0.225 ± 0.006 | 0.201 ± 0.010 | 1.322 ± 0.038 |
| Malonate | 0.190 ± 0.096 | 0.345 ± 0.081 | 0.231 ± 0.058 | 0.739 ± 0.093 | 1.882 ± 0.139 | 1.365 ± 0.102 | 0.182 ± 0.019 | 0.168 ± 0.018 | 1.378 ± 0.010 |
| Citrate + malonate | 0.227 ± 0.076 | 0.376 ± 0.081 | 0.241 ± 0.068 | 0.726 ± 0.140 | 1.951 ± 0.269 | 1.421 ± 0.224 | 0.191 ± 0.009 | 0.169 ± 0.019 | 1.377 ± 0.038 |
| Citrate | 0.228 ± 0.105 | 0.362 ± 0.097 | 0.239 ± 0.112 | 0.657 ± 0.159 | 1.951 ± 0.242 | 1.437 ± 0.175 | 0.184 ± 0.016 | 0.163 ± 0.033 | 1.358 ± 0.004 |
| ANOVA outcomes | | | | | | | | | |
| Effect of treatment | n.s.** | n.s. | n.s. | n.s. | n.s. | n.s. | **$P = 0.029$** | n.s. | n.s. |
| LSD$_{0.05}$ | | | | | | | 0.03 | | |

* Pr = pristane and Ph = phytane. **n.s. = not significant

