# Peer review of "Citrate and malonate increase microbial activity and alter microbial community composition in uncontaminated and diesel contaminated soil microcosms"

_SOIL, 2016_

## Referee Comment (RC1) · Anonymous Referee #1 · 24 Jun 2016

This article attempts to address an issue regarding phytoremediation of petroleum hydrocarbon (PHC) contaminated soils. The authors use microcosms with one or two carboxylates and test the effects on microbial community structure, activity, and PHC degradation. Hypotheses are clearly stated and the paper is straightforward and well written. However, as detailed below, the paper is extremely limited in scope, with only 1 soil and 2 carboxylates studied.

1. The authors appear to be treating this as a model system for rhizosphere activity and phytoremediation rather than a test of a simple process for remediation by direct

addition of carboxylates to contaminated soil. While the authors do discuss the limitations of the study on page 10, the experiment is so simplistic as to be of very limited value. At the very least, we would have liked to have seen, as one treatment, a more complex mixture of carboxylic acids, amino acids, and carbohydrates, all of which are substrates for microbial growth, and which would be more representative of a real rhizosphere exudate. If the authors were determined to only study carboxylates, which we see no justification for, then a wider range of carboxylates including a complex mixture of many carboxylates found in rhizosphere exudates would have been an improvement.

2. The title of the paper "Low molecular weight organic anions. . .." suggests that the results of this paper are general enough to apply to a wide range of carboxylates. But the authors only tested two carboxylates, hardly enough to generalize from. The authors should consider rewriting the title to make it more specific.

3. Only one soil was studied. Therefore, the results of this study can only be applied to that one soil. The study would be vastly improved by conducting it across a range of soil types and edaphic soil properties.

4. page 3 – soil type must be specified.

5. The concentration of diesel and of organic acids used in the experiment should be justified. Why were these concentrations chosen, and are they in any way realistic?

6. The role of pH in controlling microbial communities cannot be overstated. Two more recent references are: Lauber, C.L., Hamady, M., Knight, R., Fierer, N., 2009. Pyrosequencing-Based Assessment of Soil pH as a Predictor of Soil Bacterial Community Structure at the Continental Scale. Applied and Environmental Microbiology 75, 5111-5120. Rousk, J., Baath, E., Brookes, P.C., Lauber, C.L., Lozupone, C., Caporaso, J.G., Knight, R., Fierer, N., 2010. Soil bacterial and fungal communities across a pH gradient in an arable soil. ISME J 4, 1340-1351.

7. On page 10 you state that "further investigation is warranted. . .". While further

investigation is certainly warranted into the role of rhizosphere exudates on microbial community structure and function, including but not limited to carboxylates, it is not at all clear from the results of this paper that there is any need to continue looking at the effects of carboxylates on PHC degradation.
* * *

---

## Referee Comment (RC2) · Anonymous Referee #2 · 4 Jul 2016

General comments: In this article, the authors used microcosm experiments to determine the impact of two types of carboxylates (citrate and malonate), combined or individually, on microbial composition, activity, and degradation of petroleum hydrocarbons. Generally, the paper has been well written and very well presented through both text and illustrations. However, given that the experiments were conducted in microcosms, this article would have benefited immensely from tests on more organic compounds known to be exuded by plant roots within the rhizosphere instead of just one group. Nonetheless, it is still a useful contribution to this area of research.

[Figure]

Specific comment: The individual tests and techniques used in this study are appropriate and have been explained with a lot of details. The reader has no problem at all understanding the methodologies.

---

## Author Comment (AC1) · 21 Jul 2016

Response to referee #2

We thank the reviewer for their useful and positive comments.

You have raised a similar point as referee #1, in that a variety of compounds should have been tested (i.e. not just carboxylates). However, previous work in this area has already been conducted using a mixture of root exudate compounds (Joner and Leyval, 2003; Miya and Firestone, 2001; Xie et al., 2012). We instead attempted to focus on a single group of exudates, as these exudates have been studied extensively for their

potential role in plant nutrient acquisition and metal detoxification, but impact on the soil microbial community has received less attention (Pang et al., 2010; Pearse et al., 2006; Shane and Lambers, 2005). As far as we are aware, there are no studies examining the biodegradation of carboxylate compounds in polluted soils, and we therefore agree with you that it is still a useful contribution to this area of research.

References

Joner, E. J. and Leyval, C.: Rhizosphere gradients of polycyclic aromatic hydrocarbon (PAH) dissipation in two industrial soils and the impact of arbuscular mycorrhiza, Environ. Sci. Technol., 37(11), 2371–2375, doi:10.1021/es020196y, 2003.

Miya, R. K. and Firestone, M. K.: Enhanced phenanthrene biodegradation in soil by slender oat root exudates and root debris., J. Environ. Qual., 30(6), 1911–1918, doi:10.2134/jeq2001.1911, 2001.

Pang, J., Ryan, M. H., Tibbett, M., Cawthray, G. R., Siddique, K. H. M., Bolland, M. D. A., Denton, M. D. and Lambers, H.: Variation in morphological and physiological parameters in herbaceous perennial legumes in response to phosphorus supply, Plant Soil, 331(1), 241–255, doi:10.1007/s11104-009-0249-x, 2010.

Pearse, S. J., Veneklaas, E. J., Cawthray, G. R., Bolland, M. D. A. and Lambers, H.: Carboxylate release of wheat, canola and 11 grain legume species as affected by phosphorus status, Plant Soil, 288(1-2), 127–139, doi:10.1007/s11104-006-9099-y, 2006.

Shane, M. W. and Lambers, H.: Cluster roots: A curiosity in context, Plant Soil, 274(1-2), 101–125, doi:10.1007/s11104-004-2725-7, 2005.

Xie, X. M., Liao, M., Yang, J., Chai, J. J., Fang, S. and Wang, R. H.: Influence of root-exudates concentration on pyrene degradation and soil microbial characteristics in pyrene contaminated soil, Chemosphere, 88(10), 1190–1195, doi:10.1016/j.chemosphere.2012.03.068, 2012.

---

## Author Comment (AC2) · 21 Jul 2016

Response to referee #1

We thank the reviewers for their useful and positive comments. We have made several changes to improve the manuscript based on their suggestions. A response to each comment is provided below.

1. "The authors appear to be treating this as a model system for rhizosphere activity and phytoremediation rather than a test of a simple process for remediation by direct addition of carboxylates to contaminated soil. While the authors do discuss the

limitations of the study on page 10, the experiment is so simplistic as to be of very limited value. At the very least, we would have liked to have seen, as one treatment, a more complex mixture of carboxylic acids, amino acids, and carbohydrates, all of which are substrates for microbial growth, and which would be more representative of a real rhizosphere exudate. If the authors were determined to only study carboxylates, which we see no justification for, then a wider range of carboxylates including a complex mixture of many carboxylates found in rhizosphere exudates would have been an improvement."

Author response: We do not disagree with the referee's comments about the limitations of using a model system. However, we believe that our approach is valid for providing indications into drivers of microbial function and communities that can occur in rhizospheres. We do not attempt to test all possible experimental options, nor would this be logistically possible, rather we are taking the first steps in unraveling the mechanism involved in rhizoremediation with well-considered experimental factors. Previous work using model systems has used a combination of rhizosphere components (e.g. carboxylates in combination with carbohydrates and amino acids) (Joner and Leyval, 2003; Miya and Firestone, 2001; Xie et al., 2012). By using this approach, it is difficult to separate which component is causing the greatest shifts in microbial community structure and function. Our study is unique in that it specifically attempts to isolate the effects of two carboxylates that are commonly exuded by plant roots (Jones, 1998; Ryan et al., 2001). This is important because, whilst many studies have investigated the role of carboxylates in nutrient acquisition and aluminium detoxification by plants (Pang et al., 2010; Pearse et al., 2006; Shane and Lambers, 2005), their impact on the soil microbial community has received considerably less attention. Those studies that have examined microbial biodegradation of carboxylates have done so from a context of contrasting soil types (Fujii et al., 2010, 2012; Hashimoto, 2007; Jones and Darrah, 1994), soil horizons (Evans, 1998; van Hees et al., 2002), or in variations in the initial pH of the organic acid applied (Ström et al., 2005). As far as we are aware, there are no studies examining in detail the biodegradation of carboxylates in the context of

polluted soils, and we therefore believe that this paper makes a valuable contribution to the literature. It was beyond the capacity of this study to include every carboxylate that is potentially exuded by plant roots. Instead, we chose to focus on two of the most commonly exuded carboxylates; citrate and malonate (Jones, 1998). We also selected these carboxylates as one is a tri-carboxylate, the other a di-carboxylate and therefore expected they may provide different results due to their differing chelating abilities. It should also be noted that while a large number of carboxylates can be exuded by plant roots, typically there are only five or six present in the rhizosphere in detectable concentrations and, often, the rhizosphere carboxylate profile is dominated by only one to three carboxylates. For instance, in a study of 11 perennial legumes, native and agricultural, Pang et al. (2010) found that >95% of rhizosphere carboxylates for each species consisted of the sum of malate, malonate and citrate. Similarly, Kidd et al.(2016) found that citrate and malonate together contributed from 80% to close to 100% of the rhizosphere carboxylates in 19 out of 25 pasture legumes and grasses.

2. "The title of the paper "Low molecular weight organic anions. . .." suggests that the results of this paper are general enough to apply to a wide range of carboxylates. But the authors only tested two carboxylates, hardly enough to generalize from. The authors should consider rewriting the title to make it more specific."

Author response: We agree with the referee's comments in that the title could be misleading. We have changed the title to read "Citrate and malonate increase microbial activity and alter microbial community composition in uncontaminated and diesel contaminated soil"

3. "Only one soil was studied. Therefore, the results of this study can only be applied to that one soil. The study would be vastly improved by conducting it across a range of soil types and edaphic soil properties."

Author response: The soil selected for use in this study is representative of the majority of soil types that are present across the coastal plains of Western Australia

(Schoknecht and Pathan 2013). As this state has significant onshore oil fields, this study was aimed to be applicable to real world applications where contamination from oil pipelines is a potential issue.

4. "page 3 – soil type must be specified."

Author response: The soil type was specified in the paper as a loamy sand with 5% clay collected from the top mineral horizon. Additionally, nutrient concentrations were provided, as well as EC, pH and water retention. The pedological classification of grey chromosol (Australian soil classification) has now also been included in the paper to further define the soil.

5. "The concentration of diesel and of organic acids used in the experiment should be justified. Why were these concentrations chosen, and are they in any way realistic?"

Author response: The concentration of diesel (10,000 mg kg-1) was selected to make results comparable to previous studies on diesel bioremediation and rhizoremediation (Bento et al., 2005; Boopathy, 2004; Seklemova et al., 2001; Tesar et al., 2002). The concentrations of organic acids were selected based on concentrations that are typically found in the rhizosphere ($0.1 - 50$ $\mu$M), although the rhizosphere around cluster roots can contain concentrations in excess of 600 $\mu$M (Dinkelaker et al., 1989; Grierson, 1992; Jones, 1998; Strobel, 2001). It must be noted that obtaining reliable estimates of rhizosphere carboxylate concentrations is extremely difficult. Root concentrations are typically in the range of $10 - 20$ mM (Jones, 1998; Neumann and Römheld, 1999; Shane and Lambers, 2005). The manuscript has been updated to include the justification for using these concentrations.

6. "The role of pH in controlling microbial communities cannot be overstated. Two more recent references are: Lauber, C.L., Hamady, M., Knight, R., Fierer, N., 2009. Pyrosequencing-Based Assessment of Soil pH as a Predictor of Soil Bacterial Community Structure at the Continental Scale. Applied and Environmental Microbiology 75, 5111-5120. Rousk, J., Baath, E., Brookes, P.C., Lauber, C.L., Lozupone, C., Caporaso,

J.G., Knight, R., Fierer, N., 2010. Soil bacterial and fungal communities across a pH gradient in an arable soil. ISME J 4, 1340-1351."

Author response: We agree that pH is an important variable in influencing microbial processes and the references suggested have been included in the manuscript to further exemplify this point.

7. "On page 10 you state that "further investigation is warranted. . .". While further investigation is certainly warranted into the role of rhizosphere exudates on microbial community structure and function, including but not limited to carboxylates, it is not at all clear from the results of this paper that there is any need to continue looking at the effects of carboxylates on PHC degradation."

Author response: Although, the adding carboxylates had little direct effect on biodegradation of saturated hydrocarbons present in diesel, the microbial community did shift. As root exudates are often cited as being a major contributor to rhizoremediation and microbial functioning in the rhizosphere in general, further investigation into how these compounds may affect microbial communities in contaminated soils is warranted. Also, as the reviewer states, there is much more to learn about what might occur under different edaphic environments, hydrocarbon types and exudates.

Further references

[revised manuscript text omitted]

---

## Author Response (AR1)

**Letter to the editors regarding changes to the manuscript "Low molecular weight organic anions increase microbial activity and alter microbial community composition in uncontaminated and diesel contaminated soil"**

Dear Fuensanta García-Orenes and Lily Pereg,

We would like to thank you for your editorial comments on our research article. In light of your comments, as well as those of the referees, we have completed and resubmitted a major revision of the manuscript.

10 Specifically, we have revised the manuscript in three key areas we believe were raised during the review process:

1) **Revision of the manuscript to be presented as a preliminary study.** The manuscript has been revised to present the paper as a preliminary study on the effects of citrate and malonate on

15 bioremediation in soil microcosms instead of the previously submitted paper that made general statements on the effects of carboxylates on bioremediation. Specifically the following changes were made in this regard:

- The manuscript title has been changed to *"Citrate and malonate increase microbial activity and alter microbial community composition in uncontaminated and diesel contaminated soil*

20 *microcosms."*

- Pg 2, lines 25-33 of the introduction. Added the paragraph, *"Previous work which aimed to assess the importance of root exudates for bioremediation has often employed model systems using a combination of rhizosphere components (e.g. carboxylates in combination with carbohydrates and amino acids) (Joner and Leyval, 2003; Miya and Firestone, 2001; Xie et*

25 *al., 2012). Such an approach makes it impossible to determine which exudate component is causing the greatest shift in microbial community structure and function, and is therefore of most importance for degradation of PHCs. Carboxylate release can differ greatly among plant species with only some species, notably those species possessing cluster roots, known to*

*release high concentrations of carboxylates (Dinkelaker et al., 1989; Grierson, 1992; Jones, 1998; Strobel, 2001). To determine whether plant species with high carboxylate exudation should be prioritised when selecting plants to screen for ability to enhance PHC degradation, we devised a simple test system using two commonly exuded carboxylates."* This paragraph was added to make it clearer for the reader that this is a preliminary study with a focus on screening carboxylate-exuding plants for use in bioremediation.

- Pg 3, lines 9-15. This paragraph has been modified to highlight that the study was conducted in microcosms (and hence preliminary in nature) and to provide justification for the selection of the two carboxylates chosen in this study.

- The discussion headings and several lines throughout the discussion have been changed from *"Carboxylates…"* to *"Citrate and malonate…"* to be more specific about the outcomes of this paper rather than making general statements on the effects of all carboxylates on bioremediation.

2) **Revision of the manuscript to highlight the future research of this preliminary study.** The manuscript has been revised to provide more detail regarding the areas for follow up research. Specifically the following changes were made:

- Pg 1, lines 26-28 of the abstract. Added the sentence, *"We conclude that while the addition of citrate and malonate had little direct effect on biodegradation of saturated hydrocarbons present in diesel, their effect on the microbial community leads us to suggest further studies using a variety of soils and organic acids and linked to in situ studies of plants to investigate the role of carboxylates in microbial community dynamics."* to be more explicit about future research of this preliminary study.

- Pg 10, lines 12-16 of the discussion. Added the paragraph, *"Soil acidification poses a potential impediment to biodegradation as PHCs are optimally degraded within a near neutral pH (Dibble and Bartha, 1979). However, carboxylate exudation in plants has only been weakly correlated to rhizosphere acidification (Pang et al., 2010; Pearse et al., 2006), thus highlighting one of the limitations of applying model systems to real*

*rhizospheres and underlining the need for future in-situ studies with high carboxylate exuding plants."* This paragraph was added to highlight one limitation of the preliminary study and to provide justification for future studies *in-situ.*

- Pg 11, line 3 of discussion. Changed the header to include *'future research'.*

- Pg 11, lines 12-17 of discussion. Modified the end paragraph to *"Nonetheless, as our findings show that citrate and malonate can have a significant effect on microbial community structure in soil microcosms, further investigation focusing on plant-microbial interactions with high carboxylate exuding plants is warranted. These studies should incorporate next generation 16S sequencing technologies as well as functional gene analysis to uncover some of the key microbial players in these carboxylate-induced community shifts. Also, there is much more to learn about what might occur under different edaphic environments, with different types of PHCs and when other carboxylates or combinations of carboxylates are present."* This paragraph has been added to highlight that as this study indicated significant shifts in the microbial community with addition of the carboxylates to the soil microcosms, follow up research with in situ plants should follow.

- Pg 11. The conclusion has been modified to clearly state follow-up research to the findings of this preliminary study.

3) **Revision of the manuscript to provide clear scientific outcomes and impacts.** The manuscript has been revised to provide clearer scientific outcomes and impacts of the preliminary findings paper. Specifically the following changes were made:

- Pg 1, line 25-27 of the abstract. *"We conclude that while the addition of citrate and malonate had little direct effect on biodegradation of saturated hydrocarbons present in diesel, their effect on the microbial community leads us to suggest further studies using a variety of soils and organic acids and linked to in situ studies of plants to investigate the role of carboxylates in microbial community dynamics."* Modified sentence to be more explicit of the major findings of the paper.

- Pg 3, line 1-7 of the introduction. Added the paragraph, *"It should also be noted that while many studies have investigated the role of carboxylates in plant nutrient acquisition and aluminium detoxification (Pang et al., 2010; Pearse et al., 2006; Shane and Lambers, 2005), the impact of carboxylates on the soil microbial community has received considerably less attention. Additionally, whilst studies have examined the microbial biodegradation of carboxylates in contrasting soil types (Fujii et al., 2010, 2012; Hashimoto, 2007; Jones and Darrah, 1994), soil horizons (Evans, 1998; van Hees et al., 2002), and with variations in the initial pH of the organic acid applied (Ström et al., 2005), as far as we are aware, there are no studies examining in detail the biodegradation of carboxylates, and their effect on the microbial community, in soils contaminated with PHCs."* This paragraph was added to point out the lack of information surrounding carboxylate effects on microbial community dynamics, thus highlighting one of the major outcomes and impacts of this paper: that is, that the carboxylates caused significant shifts in community dynamics when supplied in both diesel contaminated and uncontaminated soil microcosms.
- Pg 11. The conclusion has been modified to be more explicit in stating that this preliminary study found evidence to suggest the role of carboxylates in shaping microbial communities, even in contaminated soil where one would expect hydrocarbons to have a greater effect.

The following general changes in response to specific points raised by reviewer number 1 have been made:

- Pg 3. Line 18. Both ASC and WRB soil classifications have been provided.
- Pg 3. Line 27-28. A justification for using the diesel concentration has been provided.
- Pg 4. Line 8-9. A justification for using the carboxylate concentration has been provided.
- Pg 10. Line 5. Additional references have been added on effects of pH on microbial communities.

**Tracked changes to original manuscript**

[revised manuscript text omitted]

~~In addition to enhancing microbial growth and activity, root exudates such as carboxylates, may affect microbial community function by altering either gene expression, metabolic status and/or by selecting for the growth of specific microorganisms (Benizri et al., 2002; Baudoin et al., 2003; Butler et al., 2003; Hartmann et al., 2009; Louvel et al., 2011; Yergeau et al., 2014). For example, the structure of microbial communities associated with the cluster roots of white lupin (*Lupinus albus* L.) were highly correlated to the level of carboxylate exudation (Marschner et al., 2002) and the addition of oxalate to an artificial rhizosphere environment caused more pronounced shifts in the microbial community than the addition of glucose (Keiluweit et al., 2015). The presence of PHCs is also likely to strongly influence the soil microbial community (Reddy et al., 2011; Hou et al., 2015). However, it is not known whether in a PHC-contaminated soil the addition of carboxylates will also alter microbial community composition.~~

[revised manuscript text omitted]
, on the potential effects of carboxylates on microbial community structure in soils in general and suggest that further investigation focusing on plant-microbial interactions with high carboxylate exuding plants *in-situ* is warranted. on the potential effects of carboxylates on microbial community structure in soils in general and  , particularly in regards to identifying particular microbial species or functional groups (PHC . These studies should incorporate next generation 16S sequencing technologies as well as functional gene analysis to uncover some of the key microbial players in these carboxylate-induced community shifts,  Also, there is much more to learn about what might occur

under different edaphic environments, with different types of PHCs and when other carboxylates or combinations of carboxylates are present.

**5. Conclusion**

The  key outcome of this  study  is that  citrate and malonate can stimulate microbial activity and alter microbial community structure in both uncontaminated and diesel-contaminated soils. While, we found no strong evidence to suggest that addition of citrate and malonate enhanced degradation of saturated hydrocarbons during the short (18 day) timeframe of this model system study, their impact on the microbial community leads us to suggest that further investigation of their role in plant-microbial relations is warranted using modified and more diverse experimental systems linked to *in-situ* studies of plants growing in contaminated soils.

**6. Acknowledgements**

This work was funded by an Australian Research Council linkage project (LP110201130) in partnership with Horizon Power, ChemCentre and Environmental Earth Sciences. CAP was supported by a Discovery Early Career Research Award from the Australian Research Council. Many thanks to Natalie Joyce, Daniel Ramsden, Daniel Kidd and Navjot Kaur for their technical assistance.

[revised manuscript text omitted]

* Pr = pristane and Ph = phytane. **n.s. = not significant